# Tetrazene–Characterization of Its Polymorphs

**DOI:** 10.3390/molecules26237106

**Published:** 2021-11-24

**Authors:** Jan Ryšavý, Robert Matyáš, Zdeněk Jalový, Jaroslav Maixner, Aleš Růžička, Stanislav Brandejs, Jiří Nesveda

**Affiliations:** 1Institute of Energetic Materials, Faculty of Chemical Technology, University of Pardubice, 53210 Pardubice, Czech Republic; jan.rysavy3@student.upce.cz (J.R.); zdenek.jalovy@upce.cz (Z.J.); 2Central Laboratories, University of Chemistry and Technology, Prague (UCT Prague), 16628 Prague, Czech Republic; jaroslav.maixner@vscht.cz; 3Department of General and Inorganic Chemistry, Faculty of Chemical Technology, University of Pardubice, 53210 Pardubice, Czech Republic; ales.ruzicka@upce.cz; 4Sellier & Bellot a.s., 25801 Vlašim, Czech Republic; stanislav.brandejs@sellier-bellot.cz (S.B.); nesveda@sellier-bellot.cz (J.N.)

**Keywords:** infrared spectroscopy, primers, polymorphism, Raman spectroscopy, sensitivity, tetrazene, X-ray diffraction

## Abstract

The reinvestigation of tetrazene single crystalline material by means of X-ray methods resulted in a slightly different structure when compared to previously published data. Reaction conditions responsible for different crystalline modification formation were investigated. Newly described C form was found to be the primary reaction product and the combined action of temperature and the presence of water over time is required for the transition to the A form. Both forms were described by X-ray powder diffraction. Tetrazene was also subjected to infrared and Raman spectroscopy, which allowed differentiating between the forms. The molecule was isotopically labeled with ^15^N atoms at two different locations (ring and nitrogen sidechain) and employed in assigning vibrational modes to the resulting bands. Differences between sensitivities to mechanical stimuli of the two modifications were investigated and found industrially insignificant. In the same vein, the performance of either modification in primer composition and primer was identical.

## 1. Introduction

This year marks the 100th anniversary of the first industrial use of tetrazene [1,2] and 50 years since the discovery of the correct structure [3]. Despite the substance being in use for a long time and being known even longer, much of what is now considered basic information has never been published.

Tetrazene (5-[(1*E*)-3-amidiniotetraz-1-en-1-yl]tetrazolide hydrate, known also under different names such as 1-amino-1-[(l*H*-tetrazol-5-yl)azo]guanidine hydrate and 1-amino-1-(tetrazol-5-yldiazenyl)guanidin monohydrate [4]) is one of the most industrially produced and used primary explosives (Figure 1). It is used as an energetic sensitizer in many initiating compositions, particularly in percussion and stab priming mixtures [2,5,6] used for the ignition of the powder charge in cartridges and initiation devices in other munitions products.

According to the literature, tetrazene forms two crystal modifications [3]. Crystal modification is a very important factor for explosives because it affects the density (and detonation parameters as a consequence), stability and sensitivity to external stimuli. This phenomenon is particularly well known for the primary explosive lead azide, which exists in four crystal modifications. In terms of sensitivity and stability, the only desirable and industrially produced modification is the orthorhombic alpha modification, while even a small content of other modifications in a technological product is undesirable as it increases the sensitivity of the product to mechanical stimuli and thus increases the risk of explosion during production and processing [6,7,8]. Similarly, crystal modification is an important factor in the case of two industrially produced secondary explosives. Octogen (1,3,5,7-tetranitro-1,3,5,7-tetrazocane, HMX) exists in four crystal modifications, of which only the beta modification is industrially produced. This is a result of its high density, hence high detonation parameters, and lower sensitivity to mechanical stimuli when compared to the other three modifications [9,10,11,12]. The most brisant industrially produced explosive–2,4,6,8,10,12-hexanitro-2,4,6,8,10,12-hexaazaisowurtzitane (CL-20 or HNIW)–appears in several crystal modifications where only the epsilon polymorph is produced owing to high density and absence of water of crystallization [13].

Although Duke [3] described two forms of tetrazene 50 years ago, no one has studied and characterized these modifications in detail. Therefore, we focus this study on the characterization of tetrazene forms and their impact on sensitivity and the functional parameters of tetrazene itself and the final priming mixtures.

## 2. Results and Discussion

### 2.1. Analytical Characterisation of Tetrazene Forms

#### 2.1.1. Single-Crystal X-ray

According to Duke [3], tetrazene exists in two forms–the A form and the B form, which have been recognized by X-ray analysis in his studies. However, this fact is often omitted in later works and, generally speaking, no studies have been carried out in this direction in the 50 years since the discovery.

While Duke identified his forms A and B within space groups *P2_1_/a* and *Ia*, respectively, we unambiguously found a form within space group *Cc* with unit cell parameters of a = 12.0461 Å, b = 9.3264 Å, c = 6.7419 Å, and β = 98.835° (measured at 150 K) [4]. These particular unit cell parameters and space group are found in whole temperature range of 150–293 K with only negligible changes and no phase transformation, as confirmed also by DSC measurements (more Section 2.1.4). A slight reduction of unit cell volume in comparison to Duke’s *P2_1_/a* and *Ia* forms (767 and 757 vs. 750 Å^3^) led to a slight increase in the calculated density from 1.628 and 1.651 to 1.667 g∙cm^–3^ at 150 K.

We must admit, all the interatomic distances within the molecule and between its nitrogen atoms and the oxygen of the water solvate, as well as the orientation of both molecules, are the same in our case and for form A and B (solved in *P2_1_*/*a* and *Ia* space group). It is possible to transform Duke’s *Ia* solution into our *Cc* model, but restraints for keeping good shape and main molecule composition are introduced in doing so. Ultimately, the original data are lacking in detail to support either the conjecture of B and C forms being identical or different.

For this reason, we decided to label our only slightly different form, of which we obtained a single crystal, as form C [4] in order to avoid further confusion between these forms, given their similarities. Unfortunately, A and B form single crystals were not obtained.

#### 2.1.2. X-ray Powder Diffraction

XRPD diffraction analysis was used to determine the phase composition of the powdered samples (preparation details described in Section 3.1). Based on X-ray powder diffraction (as well as on Infrared and Raman spectroscopy, see Section 2.1.3), we identified two different forms of tetrazene. The resulting powder patterns presented in Figure 2 are different and therefore contain different phases, most likely different polymorphs. The diffraction peaks are sharp, so prepared samples contain crystallites with regular 3D ordering of molecules and with the size of coherent areas in the order of micrometers.

We have performed search-match using d, I values in the Powder Diffraction Files (PDF-4+ 2021, PDF-4 Organics 2021) and have not found a proper match. The search on tetrazene has been performed in both databases as well and no hit was found. Therefore, it can be concluded that the powder data presented in this work are not available in the PDF-4 databases.

The measured powder pattern of bulk tetrazene C (preparation details described in Section 3.1) has been compared with a calculated pattern using single crystal structure data of only recently reported form C [4]. The sample of tetrazene C contains only form C. All measured diffraction peaks were indexed and are consistent with the *Cc* space group (Figure 3a).

The automatic indexing of the tetrazene C powder pattern was performed using program DICVOL04. The resulting unit cell is monoclinic with the space group *Cc* and unit-cell parameters: a = 12.070(5) Å, b = 9.328(2) Å, c = 6.815(2) Å, β = 99.210(7), unit-cell volume V = 757.3(3) Å^3^, Z = 4, T = 293 K. The figures of merits are F20 = 30.1(0.0104, 32) and M10 = 25.5. All peaks were indexed and are consistent with the *Cc* space group. The unit cell volume from the powder data is slightly larger than the unit cell volume from single crystal data: a = 12.0461(2) Å, b = 9.3264(18) Å, c = 6.7419(12) Å, β = 98.835(7), unit-cell volume V = 748.4(2) Å^3^, Z = 4, T = 150 K. This fact is caused by the thermal expansion of tetrazene. The powder data presented here were collected at ambient temperature and the single crystal experiment was performed at 150 K. On the other hand, we must admit here, the unit-cell volume calculated from indexed powder pattern parameters is exactly the same as reported for B form by Duke [3].

When the material is prepared at the recommended temperature of 56–58 °C [14], contamination of the desired C by A form, otherwise beneath the detection limit of infrared spectroscopy, is detected by the powder diffraction analysis. To obtain pure C form the reaction temperature had to be lowered to 40 °C.

The measured powder pattern of tetrazene A (sample obtained by procedure described in Section 3.1) has been compared with a pattern (Figure 3b) calculated from the unit cell parameters of form A (space group *P2_1_/a* published by Duke [3]). The tetrazene A sample contains only form A. All measured diffraction peaks were indexed and are consistent with the *P2_1_/a* space group.

The automatic indexing of the tetrazene A powder pattern was performed using program DICVOL. The resulting unit cell is monoclinic with the space group *P2_1_/a* and unit-cell parameters: a = 12.958(3) Å, b = 9.295(1) Å, c = 6.845(2) Å, β = 111.573(3)°, unit-cell volume V = 766.6(3) Å^3^, Z = 4. The figures of merits are F20 = 60.8(0.0080, 41) (20) and M20 = 30.7 (21). All peaks were indexed and are consistent with the *P2_1_/a* space group. The unit cell parameters are in perfect agreement with unit cell parameters for form A (a = 12.955(2) Å, b = 9.295(1) Å, c = 6.847(1) Å, β = 111.54(1)°, unit-cell volume V = 766.6(3) Å^3^, Z = 4) presented by Duke [3].

The indexed powder patterns of tetrazene C and tetrazene A are shown in Appendix A; the crystal data of all forms are given in Appendix A.

#### 2.1.3. Vibrational Spectra

In the tetrazene molecule, stretching vibrations of NH bonds, deformation vibrations of NH_2_ groups and several NN and CN bonds are areas of importance from a spectroscopic point of view. No strongly absorbing groups in infrared and Raman spectroscopy, such as nitro, azido or carbonyl were present. According to bond lengths determined by X-ray analysis, most of the NN bonds are between single and double bonds, the only exception being the N(1)=N(2) bond in the tetrazene moiety, which is a double bond. All CN bonds–present in tetrazole ring, tetrazene moiety and amidine moiety–are also between single and double. Most of the π electrons in the tetrazene molecule are delocalized, the tetrazole ring possesses a negative charge and the amidine moiety possesses a positive charge. The above facts make it difficult to assign the individual infrared and Raman bands to a particular moiety or part of the tetrazene molecule using general spectroscopic knowledge or spectral interpretation handbooks [15,16]. Thus, unsurprisingly, the calculated FTIR spectra of tetrazene published in [17] does not fit the real spectrum.

In order to help the band assignment, we prepared two isotopically modified molecules of tetrazene C form–abbreviated, for simplicity, to tetrazene-^15^N_1_ and tetrazene-^15^N_2_. Their structures, the exact position of the labeled nitrogen and an explanation of what “tetrazole ring” and “tetrazene chain” refer to in the spectral assignment are shown in Figure 4. Tetrazene-^15^N_1_ is labeled at the tetrazene chain and tetrazene-^15^N_2_ is labeled both at the tetrazene chain and the tetrazole ring.

By comparing the spectra of the unlabeled tetrazene with ^15^N-labeled isotopomers, several individual infrared and Raman bands can be assigned as tetrazole ring or tetrazene moiety. The rules for a decision as to whether the band is connected with the tetrazole ring or the outside of the ring (tetrazene) are as follows:If there is a shift in tetrazene-^15^N_2_ to tetrazene-^15^N_1_ and at the same time to unlabeled tetrazene, in other words, if there is a shift only in tetrazene ^15^N_2_ and no shift of ^15^N_1_ to unlabeled tetrazene, the vibration is connected with the tetrazole ring of the molecule. In Table 1, the proposed assignment is stated as “tetrazole ring”.If there is a shift in tetrazene-^15^N_2_ to unlabeled tetrazene and at the same time a shift in tetrazene-^15^N_1_ to unlabeled tetrazene, the vibration is connected with the tetrazene moiety (in other words, the linear part of the tetrazene molecule attached to the tetrazole ring). In Table 1, the proposed assignment is stated as “tetrazene chain”.

The complete infrared and Raman spectra of all tetrazene species are shown in Appendix A.

As can be seen in Table 2, in the region of 3500–2800 cm^−1^ (N–H and O–H stretching modes visible in infrared spectrum), there is no shift in FTIR bands in labeled tetrazenes. The reason is that these bands are not influenced by ^15^N atoms. Most other bands are connected or at least influenced by some ^15^N atom. The other spectral assignment was done using infrared tables and charts [15,16] and by Lieber [18], who published some infrared bands of tetrazene as a partial spectral assignment performed on the then-assumed historical structure.

X-ray analyses show little differences between the forms of tetrazene. This is reflected in FTIR and Raman spectra where all bands are located in the same areas. The largest differences are in the shape of the spectra in the region of NH stretching vibrations, visible in the infrared spectra (Figure 5), and several bands in the fingerprint region (1700–600 cm^−1^), visible both in FTIR and Raman spectra. The bands that can be used to differentiate between tetrazene forms are summarized in Table 2.

#### 2.1.4. Thermal Analysis

DSC analysis at low temperatures and DTA analysis at higher temperatures were performed to identify possible temperature-influenced phase transitions of tetrazene C.

The DSC analysis was performed by cycling the heating/cooling/heating regime in the temperature range −80 °C to 20 °C. No significant change is evident in the DSC record except for a negligible undulation of the curves at −25 °C and −50 °C (Figure 6a). The origin of this undulation is not entirely clear and is probably caused by the instrument itself. This undulation should not correspond to any phase change of tetrazene; phase change is characterised by a sharp intense peak. This result is confirmed by single crystal XRD analysis in the temperature range 150–293 K (see Section 2.1.1).

During heating of tetrazene (measured by DTA), the start of exothermic decomposition was observed at 128 °C, resulting in detonation at 137 °C. Detonation was accompanied by shattering of the measuring tube. The endo peak, which would indicate a phase transition of tetrazene or evaporation of water of crystallization, is not evident in the DTA record (Figure 6b).

### 2.2. Reaction Conditions Study

Duke states in [3] that the B form is a product of a reaction of aminoguanidium salt and sodium nitrite at temperatures below 60 °C, and the A form appears above this limit. We studied modification formation/transition, by sampling reaction mixtures over a range of conditions (initial pH 5.2–5.9 and 45–75 °C). None of our preparations led to Duke’s B form, however, we were able to obtain the A form under somewhat different conditions to those described by Duke. Instead of the reaction yielding polymorphs based on temperature limits, during any preparation procedure using the aforementioned materials, the C form is formed initially and only afterwards, depending on conditions, may it transform into the A form. This is true for this reaction regardless of the salt or conditions, be it from aminoguanidine sulfate or nitrate or aminotetrazole, led in cold or heated, thus making the C form the primary reaction product.

Besides subjecting the samples to IR spectroscopy, we also monitored the pH of the reaction mixtures. The progression of pH depends on the specific conditions of the reaction (concentration, condition of starting materials, acid used, etc.), but a general trend was observed, which can be summarized thus (Figure 7): After the initial acidification to pH 5.2–5.9, the pH rises to a little over 6 and then falls to slightly below the initial value. From the initial wave the pH increases in the first distinct steep climb (below initial value to ca. pH 6.4), followed by a slower increase, another steep climb (start at pH 7.0 to approaching final value) and finally followed by a gradual increase to the end pH value in the basic region–the limit is a little over pH 9 which, afterwards, very slowly decreases over an extended heating period.

When the reaction is carried out at the recommended temperature of 56–58 °C, tetrazene formation occurs during the pH “valley” following the initial acidification, before the reaction pH enters the climbing phase. In these conditions the production of tetrazene is finished 30 min after its initial formation and the exothermic phase is exhausted within 20 min of the initial formation, resulting in the mixture taking the temperature of the heating bath.

The first turbidity formed in the mixture corresponds to a mixture of tetrazene forms containing ca. 80% C form; however, after the reaction is finished only the C form is present (contamination by A form is beneath IR detection limit). The formation of the A form is affected by a variety of factors as is the reaction and does not correlate with any of the phases in the pH progression. Typically, after 10 h of heating, tetrazene in the mixture exists solely in the A form. However, there have been times where the transformation occurred within 5 h or took longer.

Curiously, the transformation is accompanied by a change in the product’s color to a lighter tone; this is especially noticeable when the material obtained is yellow.

A pure C form was subjected to heat stress in various conditions and the resulting mixtures were sampled to study the conditions of C-to-A form transformation. The conditions included gentle pH adjustments, concentrated buffer solutions, and both dry and wet heat.

Duke [3] stated, the formation of forms A or B during the reaction is separated by a temperature threshold. He also made a note of B transforming into A in water at ambient temperatures. While a direct comparison cannot be drawn, as his and our forms do not match exactly, considering A form as the thermodynamically stable form is correct. However, a revised statement would be, that rather than the existence of the forms being governed by temperature alone, it is the presence of water enabling the transformation of C into A to occur; elevated temperatures merely accelerate the process.

A sample of C didn′t undergo any changes within 10 days in dry heat, but in an atmosphere of 100% humidity changed into its A form (7–10 days); both at 56 °C. This process was significantly slower compared to when the transformation was carried out in a liquid medium (minutes to hours, Table 3).

The transformation rate decreases with increasing acidity of the mixture (Table 4). This effect is accentuated when higher concentrations of salts are introduced into the solution (*t*_50_ = 17 min for mildly adjusted pH 7 and 42.5 min for a concentrated pH 7 phosphate buffer). Other buffer solutions (citrate, borate) of the same pH as their respective mildly adjusted mixtures behaved similarly.

In summary, we found the speed of C-to-A form transformation to be affected by the following factors:water presence enables the transformation of C to A formspeed decreases with decreasing pH of the mixturespeed decreases with increasing concentration of dissolved salts (buffer solutions or reagents)speed increases with increasing temperature.

In addition, tetrazene can be dissolved in acids to form the corresponding salts, from which the starting material can be obtained by hydrolysis with quantitative yields [19,20]. Tetrazene recovered this way gives a predominantly C form-containing product with minor A content regardless of the starting salt or tetrazene form. The only exception known to us is the precipitation of a hydrofluoric acid solution of tetrazene by diethyl ether, which has led to the A form.

Using the preparation conditions presented in Section 3.1, bulk tetrazene forms twinned-blades (Figure 8a,b). Prolonged stirring and heating does not substantially alter the general crystal habit of the samples. The surface of the A form is rough and ragged compared to the starting state, but this can be adequately explained by partial decomposition of the material rather than the transformation itself.

### 2.3. Explosive Properties of Tetrazene Forms

#### 2.3.1. Sensitivity of Pure Tetrazene

It can be presumed, that the minute differences in the modification structures will not give rise to significant differences in stimulus sensitivities. While friction sensitivity of C and A was virtually the same, the A form was slightly more sensitive to impact than the C form. This behavior could be related to a different sensitivity of the forms. Alternatively, the difference may be explained by the comparatively rougher surface and increased porosity of the A form, which introduces additional hot spots into the material, making it more responsive to impact stimulus [21,22]. However, these differences are inconsequential for industrial applications, be it from the handling, function or safety perspective. Values of 50% activation probability of both forms are summarized in Table 5 and compared to selected standard explosives in detail in Figure 9.

#### 2.3.2. Explosive Parameters of Both Tetrazene Forms in Priming Mixtures

Tetrazene is primarily used as an energetic sensitizer for compositions in primers. Therefore, we compared functionality of both tetrazene forms in three primer mixtures using the primers force test. This test determines the pressure output of a primer mixture over time, which is important for the evaluation of priming mixture usability in primers. The pressure increase over time for NEROXIN F is presented in Figure 10a for a mixture containing tetrazene C and in Figure 10b for the same mixture with tetrazene A.

Both forms of tetrazene in NEROXIN F priming mixture give almost the same results. The only difference is the greater dispersion of NEROXIN F pressure curves with tetrazene A. This difference is probably caused by differing particles of tetrazene A resulting in an inferior quality of loading of the priming mixture into the primers when the dry filling technology is used.

The same results were obtained for NEROXIN PX and NONTOX priming mixture.

The sensitivity of priming mixture to impact was measured using the ball drop test and the results for NEROXIN F are summarized in the Table 6.

As the results of ball drop test show, the form of tetrazene does not have any noticeable influence on sensitivity of the NEROXIN F mixture. In addition, no effect of tetrazene form on impact sensitivity was observed for NEROXIN PX and NONTOX priming mixture. These results were expected and are in agreement with results of impact and friction sensitivity tests of pure tetrazene form A and C.

Results of both function parameters–pressure output and impact sensitivity–confirm that the behavior of both forms of tetrazene is virtually identical in the tested priming mixtures. The form of tetrazene, therefore, does not have an impact on the function parameters of the priming mixture.

## 3. Materials and Methods

Caution: Tetrazene is a primary explosive and must be handled with care. Safety precautions for handling explosives must be followed.

### 3.1. Synthesis

Synthesis of tetrazene A form: Tetrazene was prepared according to guidelines in [14]: Sodium nitrite (7.50 g; 109 mmol) and bisaminoguanidinium sulfate (8.75 g; 35.5 mmol) were dissolved in water (100 mL). The solution was then acidified with acetic acid at 25 °C to obtain pH 5.4 and the mixture was stirred at 56–58 °C for an extended time (see Section 2.2). After cooling to room temperature, the solid precipitate was filtered and washed gradually with water, ethanol and finally acetone, yielding 4.35 g (65.1%) of final product. *T*_ign._ = 140 °C (DTA). Anal. Calc. for C_2_H_8_N_10_O (%): C 12.77, H 4.29, N 74.44. Found: C 12.76, H 4.15, N 73.86. FTIR (cm^−1^): 3308 m, 3262 sh, 3154 m, 2977 m, 1699 m, 1625 m, 1533 m, 1482 s, 1442 w, 1413 s, 1271 m, 1202 w, 1155 s, 1104 w, 1091 m, 1080 w, 1068 s, 1039 m, 952 m, 848 sh, 807 w, 770 sh, 754 s, 725 s, 659 s. Raman (cm^−1^): 1532 w, 1497 m, 1490 m, 1441 s, 1416 s, 1203 w, 1159 w, 1101 w, 1094 w, 1073 m, 1039 vw, 956 w, 785 m, 714 vw, 614 vw, 518 w, 464 w, 426 w, 294 w. Both vibrational spectra are fully disclosed in Appendix A.

Synthesis of tetrazene C form: Tetrazene C was prepared according to the procedure of preparation tetrazene A (above), with reaction time shortened to 35 min after the precipitation is initiated. The yield of C form was 5.03 g (75.3% of theory). *T*_ign._ = 137 °C (DTA). Anal. Calc. for C_2_H_8_N_10_O (%): C 12.77, H 4.29, N 74.44. Found: C 12.47, H 4.20, N 73.69. To obtain pure C form the reaction is carried out at 40 °C for 4 h. FTIR and Raman spectra are reported in Table 1 in the Results and Discussion section and Appendix A.

Synthesis of tetrazene-^15^N_1_ C form: Isotope labeled tetrazene C form-^15^N_1_ (5-[(1*E*)-3-amidiniotetraz-1-en-1-yl](2-^15^N)tetrazolide hydrate, Figure 4) was prepared according to the modified procedure described in [24] using sodium nitrite-^15^N (Aldrich, purity 95%, 98% ^15^N): 5-aminotetrazole hydrate (0.515 g, 5.00 mmol) and aminoguanidine nitrate (0.685 g, 5.00 mmol) were dissolved in water (35 mL) at room temperature and sodium nitrite-^15^N (0.400 g, 5.70 mol) in water (3 mL) was added. A yellow turbid liquid was formed and the mixture was stirred at 24 °C for three hours. The precipitate was filtered and washed with water and ethanol yielding 0.220 g of solid (23.3%). *T*_ign._ = 135 °C (DTA). FTIR and Raman spectra are reported in Table 1 in the Results and Discussion section and Appendix A.

Synthesis of tetrazene-^15^N_2_ C form: Isotope labeled tetrazene C form-^15^N_2_ (5-[(1*E*)-3-amidinio(2-^15^N)tetraz-1-en-1-yl](2-^15^N)tetrazolide hydrate, Figure 4) was prepared via the procedure used to prepare tetrazene A which was downscaled and using sodium nitrite-^15^N (Aldrich, purity 95%, 98% ^15^N) obtained tetrazene-^15^N_2_ in A form (yield 74.5%). Form A was transformed to tetrazene-^15^N_2_ in C form by the following modified procedure described in [20,25]: Tetrazene in A form (0.056 g) was dissolved in 65% nitric acid (5 mL) and the solution was poured into water (200 mL). Dilute ammonia solution was then used to raise the pH value to 6 (precipitation of tetrazene started slowly at pH 0.5 and by 1.5 the process was complete). The resulting solid was filtered, washed gradually with water, ethanol and acetone, yielding 0.020 g (35.7%) form C tetrazene. *T*_ign._ = 132 °C (DTA). FTIR and Raman spectra are reported in Table 1 in the Results and Discussion section and Appendix A.

Tetrazene C form single crystalline material: The preparation of a single tetrazene crystal suitable for X-ray analysis was as follows: Sodium nitrite (6 g; 87.0 mmol) and bisaminoguanidinium sulfate (7 g; 28.4 mmol) were dissolved in water (100 mL). Acetic acid was used to adjust the pH to the initial value range of 5.2–5.7. The mixture was then left undisturbed at laboratory temperature. The following day large orange-yellow crystals were found (Figure 8c). The mixture remains active and after a week further product can be obtained; at least a total of 3.5 g (70.7%) of product is obtainable.

### 3.2. Measuring Tmethods

pH measurement: The reaction mixture pH was measured using Schott Instruments Lab 850 pH meter (Mainz, Germany) equipped with SI Analytics pH electrode BlueLine 14 pH (Mainz, Germany). Fisher Scientific pH 4.00, 7.00 and 10.00 buffer solutions were used for three-point calibration of the electrode.

DTA: Differential thermal analysis was carried out with a DTA 550 Ex thermal analyzer produced by OZM Research (Hrochův Týnec, Czech Republic). The samples were tested in open glass micro-test tubes in contact with air. The weight of samples was 3–5 mg, the heating rate was 5 °C∙min^−1^. Decomposition of tetrazene was accompanied by a strong acoustic effect and the destruction of the micro-test tube.

DSC: Differential scanning calorimetry was carried out with a DSC thermal analyzer DSC Q2000 produced by TA Instruments (New Castle, DE, USA). Measurements were performed at a heating/cooling rate of 20 °C∙min^–1^ in closed aluminium sample pans with a hole on the top for gas release, with a nitrogen flow of 50 mL∙min^–1^.

Elemental analysis: it was carried out using automatic elemental analyzer UNICUBE (Elementar, Langenselbold, Germany) on 1–2 mg samples.

FTIR: Infrared spectra were collected using a Nicolet iS50 FT-IR spectrometer (Thermo, Madison, WI, USA) with an ATR single reflection ZnSe accessory GladiATR (PIKE, Fitchburg, WI, USA). Measurement parameters were: spectral region 4000–600 cm^−1^, resolution 4 cm^−1^ and number of scans 64 (for isotopically labeled materials the resolution was 1 cm^−1^ and number of scans 128).

Raman spectroscopy: Raman spectra were measured using the Nicolet iS50 Raman module. Excitation laser 1064 nm, power of laser 50–500 mW, with defocusing lens to avoid rapid sample decomposition, spectral region 4000–100 cm^−1^, resolution 4 cm^−1^ and number of scans 96. Spectral manipulation for both, infrared and Raman, spectra was done using Omnic 9.2 software.

X-ray analysis: Full-sets of diffraction data for tetrazene were collected at 150(2) K and 293(2) K, respectively, with a Bruker D8-Venture diffractometer equipped with Mo (Mo/Kα radiation; λ = 0.71073 Å) microfocus X-ray (IµS) sources, Photon CMOS detector and Oxford Cryosystems cooling device was used for data collection. The frames were integrated with the Bruker SAINT software package using a narrow-frame algorithm. Data were corrected for absorption effects using the Multi-Scan method (SADABS). Resulting data were treated by XT-version 2014/5 and SHELXL-2017/1 software implemented in APEX3 v2016.5-0 (Bruker AXS) system [26]. Hydrogen atoms were localized on a difference Fourier map. Crystallographic data for structural analysis have been deposited with the Cambridge Crystallographic Data Centre, CCDC no. 2103722. Copies of this information may be obtained free of charge from The Director, CCDC, 12 Union Road, Cambridge CB2 1EY, UK (fax: +44-1223-336033; e-mail: deposit@ccdc.cam.ac.uk or www: http://www.ccdc.cam.ac.uk, accessed on 17 August 2021).

The diffraction data were collected at room temperature with an X’Pert3 Powder *θ-θ* powder diffractometer with parafocusing Bragg-Brentano geometry using Cu *Kα* radiation (*λ* = 1.5418 Å, Ni filter, generator setting: 40 kV, 30 mA). An ultrafast PIXCEL detector with 255 channels was employed to collect XRD data over the angular range from 5 to 80° 2*θ* with a step size of 0.026° 2*θ* and a counting time of 0.618 s/step. The software package HighScore Plus V 4.8 (PANalytical, Almelo, Netherlands) was used to smooth the data, to fit the background, to eliminate the *Kα_2_* component, and the top of the smoothed peaks were used to determine the 2*θ* peak positions and intensities (I-values) of the diffraction peaks. The d-values were calculated using the Bragg law and Cu *Kα_1_* radiation (*λ* = 1.5406 Å).

SEM: Samples of tetrazene were visualized with a scanning electron microscope (SEM; Jeol JSM 5500 LV) (JEOL, Akishima, Japan).

Sensitivity measurement: Sensitivity to friction was determined using small BAM apparatus type FSA-12. The testing set consisted of porcelain BFST Pt 100 25 × 25 mm plates and BFST Pn 200 pegs. Sensitivity to impact was measured using BAM fall hammer. Testing sets composed of steel guides BFH-SC and cylinders BFH-SR. All sensitivity measurement apparatus and related supplies were manufactured by OZM Research (Hrochův Týnec, Czech Republic). Sensitivities to friction and impact were evaluated using probit analysis on 15 trials on each intensity level (at least five levels where possible) and results were expressed as a friction force or impact energy with 50% probability of initiation [27].

Performance of priming mixtures: Priming mixtures containing tetrazene A or C and functionality of final primers was determined in primers force test (Lachaussée s.a. type 765/60-64). Priming mixtures NEROXIN F (containing lead styphnate/tetrazene/Ba(NO_3_)_2_/Sb_2_S_3_/PbO_2_/Pb_3_O_4_), NEROXIN PX (containing lead styphnate/tetrazene /Ba(NO_3_)_2_/Sb_2_S_3_/PbO_2_/Pb_3_O_4_/pentaerythritol tetranitrate) and NONTOX (tetrazene/pentaerythritol tetranitrate/KNO_3_/boron/ glass powder/nitrocellulose) were prepared using standard production technology from the Sellier&Bellot company for both forms of tetrazene. The primer type 4.4 SP was used for NEROXIN F composition, type 5.3 LR for NEROXIN PX and type 4.4 SP Nontox for NONTOX composition. The completed primers were fitted into closed manometric vessels containing dynamic pressure sensor model 102B03 (PCB Piezotronics Inc.). The primers were initiated by the fall of a 112 g ball from a height of 30 cm onto a striker which hits the primer and the charge output pressure vs. time characteristic was measured (measurement range was from atmospheric pressure to 39.6 MPa); 20 runs were performed per sample.

Sensitivity of primers: Sensitivity of primers to impact was measured using the same apparatus that was used for performance measuring of the priming mixture. Primers were initiated by the fall of a 112 g ball onto a striker with primer.

## 4. Conclusions

Tetrazene is a widely used industrial primary explosive which finds its use as an energetic sensitizer in primer and initiator compositions. Although the compound has been known and used for a long time, many of tetrazene’s important properties have never been described or published.

The existence of two forms of tetrazene was confirmed, although our results do not quite match those previously published by Duke. Tetrazene C form within the space group *Cc* with calculated density of 1.667 g∙cm^−3^ forms as a result of industrial production. In addition, it was found that the C form within the tetrazene reaction mixture slowly transforms into the A form after several hours. The rate of this transformation depends on pH–lower mixture pH inhibits the transition. The transition to A form occurs even in dry C form, when tetrazene is exposed to a moist atmosphere at elevated temperatures.

Infrared spectrum bands were partially assigned with the help of variously ^15^N labeled tetrazene.

The A form is more sensitive to impact and friction than the C form, but the difference in sensitivities between the modifications is very small. Both tetrazene forms were tested in three primer compositions using primers force test. This test determines the pressure output of primer mixture over time, which is important for the evaluation of priming mixture usability in primers. Both tetrazene forms performed equally in this test. The difference in impact sensitivity of the three primer compositions, with regards to the tetrazene form used, is negligible. Thus, from the application point of view the tetrazene form used should not affect the function parameters of the primer. This in turn is important for producers to know that the accidental use of A form in the product does not raise safety or function concerns in the end product.

## Figures and Tables

**Figure 1 molecules-26-07106-f001:**
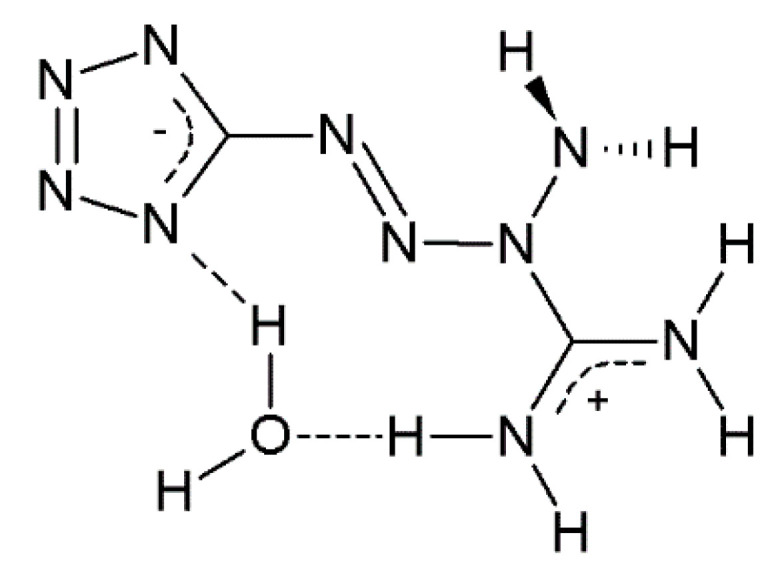
Chemical structure of tetrazene.

**Figure 2 molecules-26-07106-f002:**
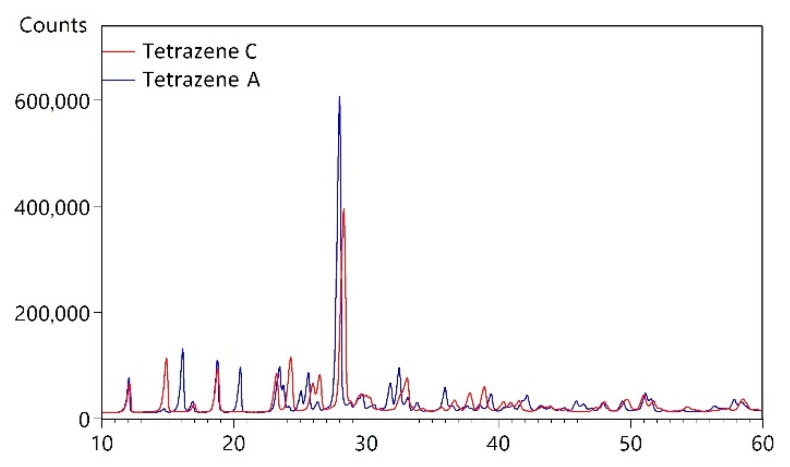
X-ray powder diffraction pattern of tetrazene A and C.

**Figure 3 molecules-26-07106-f003:**
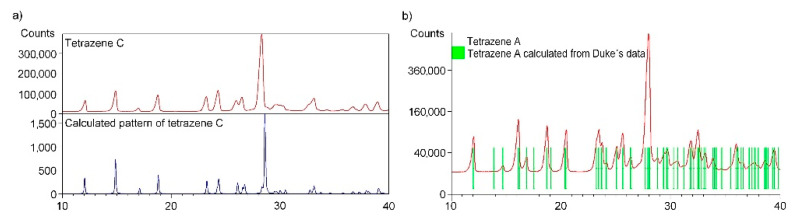
Comparison of powder diffraction pattern of tetrazene C with a calculated pattern using single crystal structure data for form C (**a**); comparison of powder diffraction pattern of tetrazene A with a calculated pattern using Duke’s data [3] for form A (**b**).

**Figure 4 molecules-26-07106-f004:**
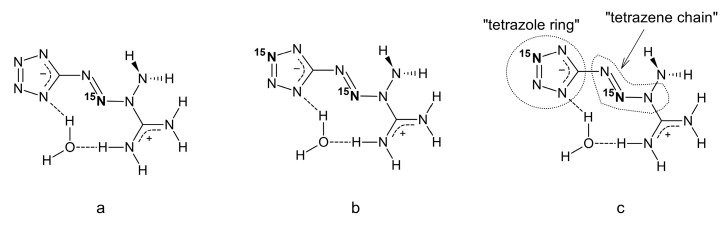
Structural formulas of ^15^N_1_ labeled tetrazene (**a**); ^15^N_2_ labelled tetrazene (**b**) and explanation of terms used in infrared spectra assignment (**c**).

**Figure 5 molecules-26-07106-f005:**
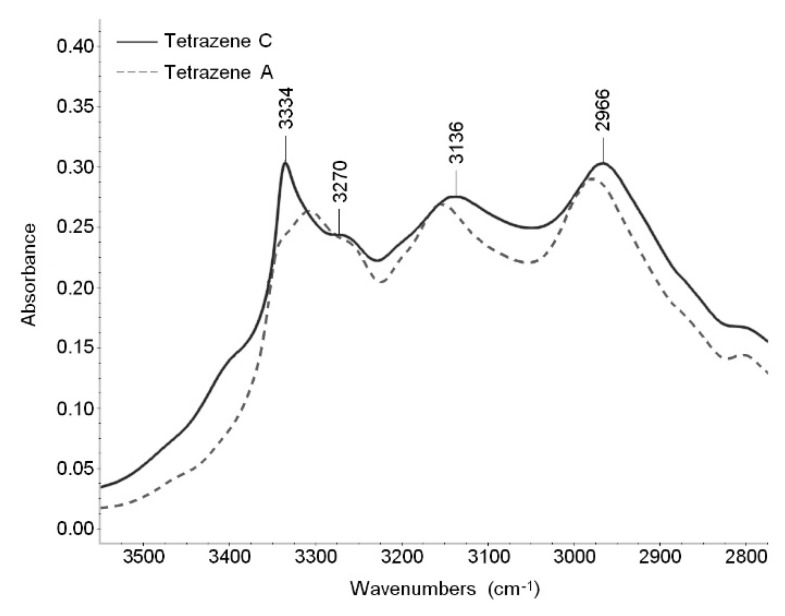
Infrared spectra of C and A form of tetrazene (region 3500–2800 cm^−1^).

**Figure 6 molecules-26-07106-f006:**
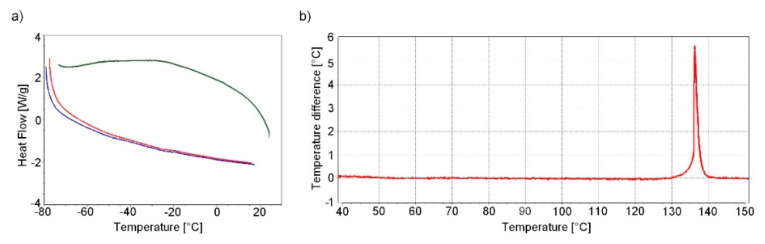
DSC record of tetrazene C at low temperatures (measurement mode was used: start from −80 °C to 20 °C (red curve)–cooling to −80 °C (green curve)–heating to 20 °C (blue curve) (**a**); DTA record of tetrazene C during heating (**b**)).

**Figure 7 molecules-26-07106-f007:**
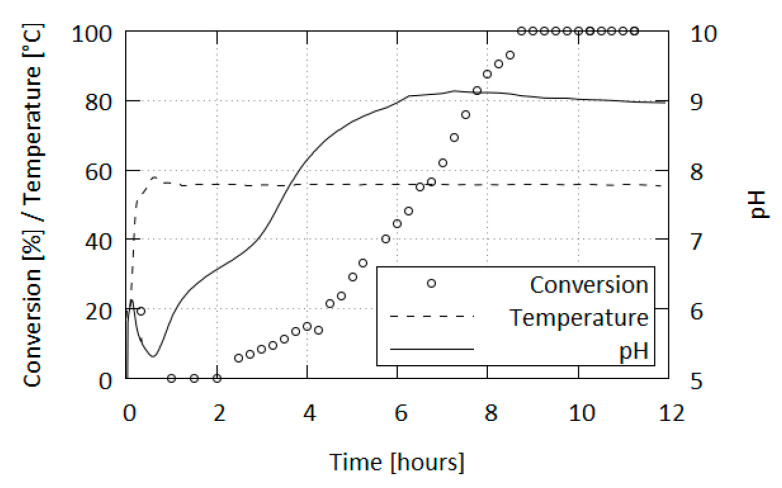
Progression of form conversion (C to A), reaction temperature and pH over reaction time.

**Figure 8 molecules-26-07106-f008:**
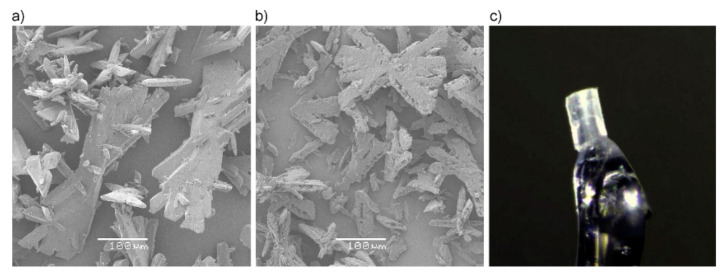
SEM images of tetrazene C (**a**) and A (**b**), both at 200× magnification and single crystal used for XRD analysis (**c**).

**Figure 9 molecules-26-07106-f009:**
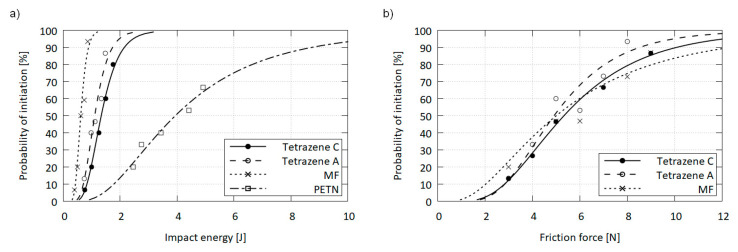
Impact (**a**) and friction (**b**) sensitivity curves for tetrazenes compared with mercury fulminate (MF) and pentaerythritol tetranitrate (PETN).

**Figure 10 molecules-26-07106-f010:**
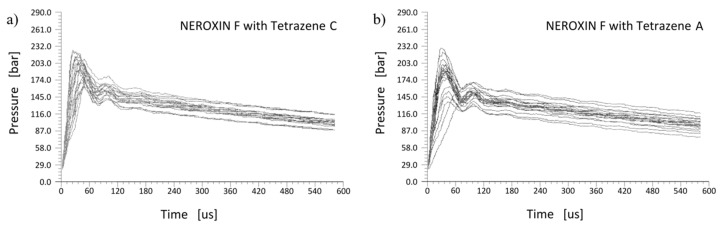
The pressure output over time in the primers force test for NEROXIN F with tetrazene C (**a**) and A (**b**); each graph contains results for 20 trials.

**Table 1 molecules-26-07106-t001:** Vibrational frequencies of tetrazene and ^15^N labeled tetrazenes of C form.

FTIR [cm^−1^]			Raman [cm^−1^]			Proposed Assignment
Tetrazene	Tetrazene-^15^N_1_	Tetrazene-^15^N_2_	Tetrazene	Tetrazene-^15^N_1_	Tetrazene-^15^N_2_	
3334 m	3335 m	3335 m				NH str.
3270 m	3270 m	3270 m				NH str.
3136 m	3135 m	3134 m				NH str.
2966 m	2965 m	2967 m				NH str.
1699 m	1699 m	1699 m				CN (in amidine)
1623 m	1623 m	1618 m				Tetrazole ring
1536 m	1529 m	1529 m	1534 m	1532 m	1532 m	Tetrazene chain N=N str.
1491 s	1471 s	1468 s	1501 s	1478 s	1478 s	Tetrazene chain N=N str.
1440 m	1440 m	1438 m	1440 s	1439 s	1438 s	Tetrazole ring or C(tetrazole)–N
1415 m	1414 m	1412 m	1421 m	1418 m	1417 m	Tetrazene chain NN str.
1269 m	1269 m	1270 m				NCN o. ph. or NH_2_ rock.
1201 w	1199 w	1192 w	1201 w	1198	1191	Tetrazole ring
1154 s	1151 s	1148 s	1158 m	1153	1150	Tetrazene chain N–N str.
1104 m	1104 m	1102 m	1099 vw	1099	1099	C–N or NH_2_ rock.
			1092 m	1092	1084	Tetrazole ring
1080 s	1079 s	1073 s				Tetrazole ring
1069 s	1069 s	1066 s	1071 m	1070	1066	Tetrazole ring
1038 m	1039 m	1031 m	1039 vw	1039	1032	Tetrazole ring
952 m	943 m	942 m	955 w	945	946	Tetrazene chain N–N str. or NN bend.
831 m	833 m	835 m				?
770 sh	770 sh	770 sh				?
			784 m	781	782	Tetrazene chain N–N str. or NN bend
756 s	755 s	755 s				NH_2_ wagg
727 m	728 m	721 m				Tetrazole ring
687 s	689 s	685 s				Tetrazole ring or NH_2_ rock. ?
624 w	620 w	620 w	615 w	610 w	612 w	Tetrazene chain NN bend.
			520 w	518 w	517 w	Tetrazene chain (def.)

Abbreviations used: s—strong, m—medium, w—weak (intensities), sh—shoulder, str—stretching, def. —deformation vibration, bend.—bending, o. ph.—out of phase, rock.—rocking, wagg.—wagging, ?—undecided.

**Table 2 molecules-26-07106-t002:** Vibrational bands useful in C and A tetrazene form differentiation.

Infrared [cm^–1^]		Raman [cm^–1^]	
C Form	A Form	C Form	A Form
3334 m	3308 m	1501 s	1497 m, 1490 m
3136 w	3154 m	1421 m	1416 s
2966 m	2977 m	1071 m	1073 m
1623 m	1625 m	291 w	294 w
1491 s	1482 s		
1415 m	1413 s		
1269	1271		
831 m	807 w		
727 m	725 s		
687 s	659 s		
624 w	-		

**Table 3 molecules-26-07106-t003:** Temperature dependence of conversion half-time of C to A form in aqueous environment.

Temperature [°C]	45	56	65
*t*_50_ [min]	153	31.5	6.5

**Table 4 molecules-26-07106-t004:** pH dependence of form C-to-A conversion half-time.

Sample	pH 5	pH 6	pH 7	pH 8
*t*_50_ [min]	32	28	17	15.5

**Table 5 molecules-26-07106-t005:** Friction and impact sensitivity at 50% initiation probability.

	*E*_50_ [J]	*F*_50_ [N]	Ref.
GNGT–C	1.34	5.14	
GNGT–A	1.13	4.88	
Hg(CNO)_2_	0.62	5.29	[23]
PETN–type NS	3.93	75.1	[23]

**Table 6 molecules-26-07106-t006:** Results of the ball drop test for NEROXIN F with tetrazene C and A forms.

Drop	Ratio Activation/No Activation for NEROXIN F	
[mm]	with Tetrazene C	with Tetrazene A
175	25/0	-
150	24/1	25/0
125	23/2	22/3
100	9/16	7/18
75	0/25	0/25

## Data Availability

The data presented in this study are available in the article and the Appendix A.

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
