# Peer review of "Tetrazene–Characterization of Its Polymorphs"

_molecules, 2021, doi:10.3390/molecules26237106_

Round 1
Reviewer 1 Report
- I’have no time to do it, so I ask the authors to doubly check the name reported at page 1, line 39. (5-[(1E)-3-amidiniotetraz-1-en-1-yl]tetrazolate hydrate); is it the actual correct IUPAC name for the specific molecule they studied? And/or has to be further enriched with a more common name.
- Figure 3 at pag. 4: the bottom green plot is totally misleading. The paper of Duke (J. Chem. Soc. D 1971) report the unit cells of two possible polymorphs, but not the full crystal structure of them, in particular atom coordinates. Although several crystallographic routines plot XRD powder patterns simply giving cell parameters and space group number, it is completely arbitrary. The peak intensity is not achievable without the full set of atom positions. The green plot has to be replaced by vertical bars that give simply peak positions. In my opinion Figures 2 and 3 should be merged in a unique plot.
- Single crystal data: although in 'Materials and Methods' section the analytical procedure is described, the structure of ‘polymorph C’ is not reported. The *.pdf file should be deposited in CCDC database and adequately referenced, or reported as Supplementary Information, or if the Authors have used it as separate publication, it must be clearly referenced (in that case this paper will lose in novelty).
Reviewer 2 Report
This paper reports the structure of a new polymorphic crystal of tetrazene, guanylnitrosaminoguanyltetrazene (1-(5-tetrazolyl)-4-guanyltetrazene hydrate), a substance used in explosives, and its characterization by XRD, FT-IR, Raman spectroscopy and related properties are reported. I thought it was very interesting that the crystal structure of the new polymorphic crystal of tetrazene was revealed. It is also important that this manuscript reports the crystallographically important properties such as polymorphic transition and morphology of the crystals.
However, the authors do not describe the crystal structure of polymorph C other than the lattice parameter. In the case of single crystal structure analysis, the crystal structure (CIF) should be deposite to CCDC and the deposite number should be obtained and included in the manuscript. It is very important for this paper that the crystal structure is available to the reader and that the structure is clarified and discussed. Unfortunately, CIF is not available, so this review is difficult.
From Figs. 1, 2, and 3, I believe that the authors have obtained a novel polymorph C. But, because the crystal structure has not been disclosed, I cannot say for sure. The XRD patterns are not enough for new crystal structure. Also, when XRD was indexed, an indexed XRD is necessary to show the result.
The title of this paper should be "Tetrazene - characterization of its new polymorph".
Duke described the molecular structure in the paper. In this paper, the authors state that polymorph C has approximately the same molecular structure, but the structure should be shown and discussed in the paper.Also, the The ORTEP diagram and packing diagram are needed.
The authors are comparing unit lattice volumes in L80. Since lattice constants are dependent on the temperature at which they are measured, the lattice constants used in the comparison and the temperature at which they were measured should be stated. the lattice constants for L76 are either from XRD or from single crystal structure analysis (150K).
Similarly, I think the density of the crystals depends on the temperature; you compare the densities in L81, but the value for polymorph C should include the measured temperature. Perhaps you are using the lattice volume obtained from XRD of polymorph C.
L78, and report the DSC results. I think you need a DSC chart in the appendix. I think the reader is interested in whether DSC or TG/DTA shows phase transition or dehydration summation. Is it possible to measure the range of temperatures higher than room temperature?
The text in L252 to 253 is hard to understand; what did Duke actually say about polymorph C? Duke said that polymorph A is the thermodynamically most stable phase. The results of this study also support this, so it should be added to the conclusion.
Reviewer 3 Report
This manuscript describes a polymorph of tetracene. The work appears to be written rushed, with guiding sentences of the manuscript template still in place (chapter 2, 'This section may be divided ... ').
When talking about polymorphs one would like to see packing diagrams to show the differences between the polymorphs/modifications, which are not given. The authors should comment weather the previous cell can be transformed into that of the new found polymorph. A table with the lattice parameters of all polymorphs/modifications and chemical moiety formulas should be given.
In a revision, the cif-file and checkcif result for the single crystal structure needs to be submitted as well for a more thorough investigation.
Reviewer 4 Report
Tetrazene - characterization of its polymoprhs
This paper describes the formation of a new polymorph/solvate of tetrazene, its characterisation and its activity as an explosive compared to the known form A. The authors have carried out an extensive study of the material, however, the paper lacks clarity and cannot be accepted in its present form. Major revisions addressing the following points needs to be carried out before it can be considered for publication.
- The authors use the term ‘crystal modification’ extensively in the paper, while the title clearly says polymorph. However, on reading the paper it seems to be hydrate form. This should be clarified and the correct terminology must be used.
- The Methods sections says that the preparation of Form C is described in section 2.1 however, this is not the case. The preparation of the main material seems to be missing.
- The first few sentences in the results and discussion section seems to be the generic text from the manuscript template. This should be removed.
- While the authors describe the single crystal experiments and DSC, the CIF is not presented, neither is the DSC data. A better comparison of the structural aspects should be made with Form B as well as it is the hydrate form.
- Figures 1,2 and 3 can be combined into subplots as they all describe PXRD data.
- Line 129 again says sample obtained by procedure 2.1 – which is missing.
- It will be best to move Fig 4 to the beginning so that the readers can see what the molecule looks like in the beginning, and the labelling of “tetrazole ring” and “tetrazene chain” is a bit confusing as the molecule is fairly large, perhaps these can be highlighted in the diagram as well for clarity.
- Line 198 has a typo, the ‘d’ from infrared is missing.
- There is a lot of interesting spectroscopic data mentioned here with elaborate assignment tables, however, the spectra themselves have not been presented. This should be made available at least in the supporting information, for both IR and Raman.
- Were the crystal structures or the PXRD of the isotope labelled versions examined? If so these should be presented in the supporting information as well.
- The actual pictures of the yellow-orange crystals must be presented.
- Line 238 states, “ Typically, after 10 hours of heating, tetrazene in the mixture exists soley in the A form. However, there have been times where the transformation occurred within 5 hours or took longer.” This is slightly worrying. Is there a reason behind this behaviour?
- The description of the crystal habits as “X”, “Y” and “V” is a bit colloquial, wouldn’t it be more accurate to describe the crystals in the images as “twinned – blades” or aggregate of blades? If they are indeed twinned – was there an effect on the single crystal analysis?
- Figures 8 and 9 can be made into one figure with subplots, as can Figure 10 and 11.
- Figure 10 and 11 should be coloured to describe the change in pressure for the 20 different runs as Form A seems to be quite different under certain pressures, but this is not clear from the black and white plots.
These changes must be carried out to make the paper more legible and the some grammatical corrections of the language needs to be carried out as well.
Round 2
Reviewer 1 Report
Even the novelty of the paper is decreased because the presence of a parallel paper (ok with a different target, but with some overlaps) the quality of presentation is growth up.
I suggest the publication in the present form.
Reviewer 2 Report
The authors have taken into account the reviewers' opinions and have revised the manuscript well. The supplementary material has been well prepared.
There is one point that concerns me. As described by the authors (P4 2.1.2), the lattice parameter of Duke's polymorph B is the same as that of the polymorph C named in this study. We cannot compare PXRD between polymorphs B and C because Duke does not disclose the atomic coordinates in his [3] paper. However, Duke's Ia cell can be transformed into a Cc cell (12.058 9.332 6.811 90 98.99 90) using the transformation matrix of (1, 0, 1; 0, -1, 0; 0 0 -1). This lattice constant can be regarded as identical to polymorph C. Therefore, the authors may have avoided naming it polymorph C and used polymorph B instead.
Therefore, in the paper, the authors should have specifically described the lattice transformation above and clearly stated that polymorph C is likely to be the same phase as polymoprh B, given the lattice constants.
Since Duke [3] has not disclosed the atomic coordinates, I think it is very important that the crystal structure of form C (or B) was analyzed in this study and included in the CSD. We hope that the crystal structure of form A will be analyzed in the future.
As a small matter, the DOI of Ref [4] is wrong. Please check and change it to 10.1021/acs.cgd.c01044.
Reviewer 3 Report
Although I do not favor one scientific result being exploited in two different papers that easily could be combined into one, I think the authors did well implementing the reviewer's recommendations. The single X-ray structure has now been supplied and checkcif did not detect significant problems.
Author Response
Reviewer comment: English language and style are fine/minor spell check required.
Our answer: On your recommendation, we have revised the English language again.